

# Concentration of quantum equilibration and an estimate of the recurrence time

Jonathon Riddell[1,2*], Nathan J. Pagliaroli[3†] and Álvaro M. Alhambra[4‡]

**1** Department of Physics & Astronomy, McMaster University,
1280 Main St. W., Hamilton ON L8S 4M1, Canada
**2** Perimeter Institute for Theoretical Physics, Waterloo, ON N2L 2Y5, Canada
**3** Department of Mathematics, Western University,
1151 Richmond St, London ON N6A 3K7, Canada
**4** Max-Planck-Institut für Quantenoptik, Hans-Kopfermann-Straße 1,
D-85748 Garching, Germany

★ jonathon.riddell@nottingham.ac.uk , † npagliar@uwo.ca , ‡ alvaro.alhambra@csic.es

## Abstract

We show that the dynamics of generic quantum systems concentrate around their equilibrium value when measuring at arbitrary times. This means that the probability of finding such values away from that equilibrium is exponentially suppressed, with a decay rate given by the effective dimension. Our result allows us to place a lower bound on the recurrence time of quantum systems, since recurrences corresponds to the rare events of finding a state away from equilibrium. In many-body systems, this bound is doubly exponential in system size. We also show corresponding results for free fermions, which display a weaker concentration and earlier recurrences.

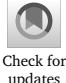

# 1  Introduction

Closed quantum systems obey the Schrödinger equation, so that their dynamics are both unitary and reversible. Most large systems, however, seem to quickly evolve towards a steady state for long times, with only very small out-of-equilibrium fluctuations around it. This process is usually called *equilibration*, and is associated with the emergence of statistical physics [1, 2]. The equilibrated or average expectation value of an observable $A$ is

$$\overline{\langle A \rangle} = \lim_{T \to \infty} \int_0^T \frac{\mathrm{d}t}{T} \langle A(t) \rangle \,, \tag{1}$$

where $\langle A(t) \rangle = \langle \Psi | e^{-iHt} A e^{iHt} | \Psi \rangle$ for some initial state $\Psi$ and Hamiltonian $H$.

If a system equilibrates, it is because the probability of finding $\langle A(t) \rangle$ very close to $\overline{\langle A \rangle}$ at any given time is overwhelmingly large. We show that this is indeed the case: the dynamics of quantum systems with a *generic* spectrum concentrate highly around the steady-state value $\overline{\langle A \rangle}$.

More specifically, we show that when sampling times at random $t \in [0, \infty)$ the probability of finding the system away from equilibrium is exponentially suppressed. The decay rate of that exponential is given by the *effective dimension* or *inverse participation ratio*. This is defined as $\mathrm{Tr}\left[ \omega^2 \right]^{-1}$, where $\omega$ is the diagonal ensemble

$$\omega = \lim_{T \to \infty} \int_0^T \frac{\mathrm{d}t}{T} e^{-iHt} | \Psi \rangle \langle \Psi | e^{iHt} \,, \tag{2}$$

which is such that $\mathrm{Tr}[A\omega] = \overline{\langle A \rangle}$. A similar result to what we here prove was argued to hold as a consequence of the ETH in [3].

That this equilibration happens, leaving little or no memory from the initial conditions, seems to conflict with the unitarity and reversibility of the dynamics. This conflict can be seen by considering the Poincaré recurrence theorem in quantum mechanics [4–8], which states that any closed quantum evolution eventually returns arbitrarily close to its initial state.

The solution to this problem is that even if the initial state is eventually recovered to an arbitrarily good approximation, this only happens at extremely long times. These recurrences constitute large out-of-equilibrium fluctuations, that can be understood as the rare events of finding a system far from its equilibrated state.

Based on this idea, we show how a lower bound on the average spacing between recurrences follows from our concentration results, as the inverse of the tail bound. We find that recurrences occur at time intervals that are at least exponential in the effective dimension.

Table 1: Lower bounds on the recurrence time for different dynamical quantities. $\mathrm{Tr}\left[\omega^2\right]^{-1}$ is the effective dimension of a system with generic spectrum and $L$ is the number of sites in a fermionic lattice. In the free case, the observable and initial state are restricted to specific forms. See below for the precise statements.

|  | $\langle A(t)\rangle$ | $|\langle\Psi| e^{-itH} |\Psi\rangle|^2$ |
|---|---|---|
| Generic | $e^{\Omega\left(\mathrm{Tr}\left[\omega^2\right]^{-1/2}\right)}$ | $e^{\Omega\left(\mathrm{Tr}\left[\omega^2\right]^{-1}\right)}$ |
| Free | $e^{\Omega(\sqrt{L})}$ | $e^{\Omega(L)}$ |

This gives a mathematically rigorous scaling on the average recurrence time, that matches the scaling of previous estimates [9] and exact calculations [10]. See [11, 12] for other related results.

We also show equivalent results for free fermion Hamiltonians with generic single-particle modes. We find that under the assumption of extensivity in the single particle eigenstates a similar concentration bound and recurrence time result hold, but with a slower exponential scaling on the lattice size. This shows the markedly different behaviour with respect to generic models. See Table 1 for a summary.

Our results constitute a qualitative improvement over previous bounds on out-of-equilibrium fluctuations [13–15] for systems with a generic spectrum. These only focused on the variance induced by the probability measure $\lim_{T\to\infty}\int_0^T\frac{dt}{T}$, while we are able to analyze arbitrarily high moments thereof. The improvement is exponential in the same sense in which the Chernoff-Hoeffding bound is exponentially better than Chebyshev's inequality.

## 2 Concentration bound

We consider functions of time $f(t)$ that track some physical property of interest. In the cases here, $f(t) = \langle A(t)\rangle$ is the expectation value of a time-evolved operator $A(t)$. This allows us to define the moments of a probability distribution

$$\overline{f} \equiv \lim_{T\to\infty}\int_0^T \frac{dt}{T} f(t), \tag{3}$$

$$\mu_q \equiv \lim_{T\to\infty}\int_0^T \frac{dt}{T}\left(f(t)-\overline{f}\right)^q. \tag{4}$$

These moments are bounded for arbitrary $q$ as $\mu_q \leq (2\|A\|)^q$. This means that they uniquely determine a characteristic function with an infinite radius of convergence

$$\phi(\lambda) = \sum_q \frac{\mu_q\lambda}{q!}. \tag{5}$$

This function defines a probability distribution, which we can write formally as

$$P(x) = \lim_{T\to\infty}\int_0^T \frac{dt}{T}\delta(x - f(t)). \tag{6}$$

Here $P(x)$ should be understood as the probability that, if we pick a random time $t \in [0,\infty)$, the value of $f(t)$ is exactly $x$ (see also [16] and [17] for an overview of previous results).

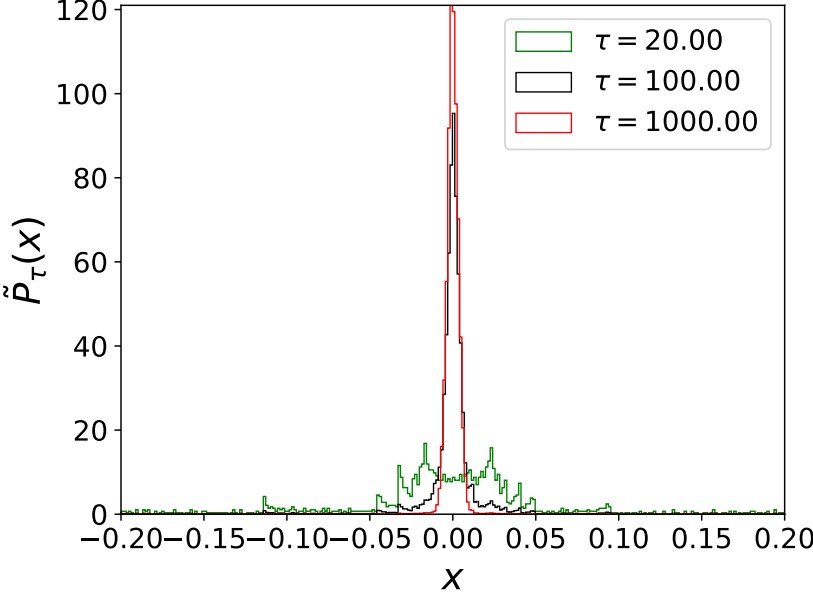

Figure 1: Convergence of $P_T(x)$ to $P(x)$ where $P_T(x) = \int_0^T \frac{dt}{T} \delta(x - \langle A(t) \rangle)$ and $P(x)$ is recovered as $T \to \infty$. $\tilde{P}_T(x)$ represents the approximation of $P_T(x)$ by binning samples and constructing a histogram. 1000 bins were used to create this histogram. Numerics were performed on a spin $1/2$ chain with 22 sites. The data is normalized such that $\int_{-\infty}^{\infty} \tilde{P}_T(x) dx = 1$. The Hamiltonian is a Heisenberg type model with nearest and next nearest neighbour interactions. The parameters in the Hamiltonian are chosen so that we have a non-integrable model. Further details can be found in App. D.

Note that in order to compute $\int x^q P(x) dx$ the limit in $T$ is swapped with the integral in $x$. For justification of this see Appendix A. An example of $P(x)$ forming for $f(t) = \langle A(t) \rangle$ is given in Fig. 1. Fig. 1 shows how averaging over a finite interval gives concentrated distributions around equilibrium. The distribution becomes sharper as we increase the length of time being averaged over.

Below we prove that the moments $\mu_q$ are bounded by

$$\kappa_q \leq (qg)^q \,, \tag{7}$$

where $g$ is some small quantity, decreasing quickly as the size of the system grows and such that $g \to 0$ in the thermodynamic limit. A bound of this form implies that the distribution concentrates highly around the average, as per the following elementary lemma.

**Lemma 1.** *Let $|\mu_q| \leq (qg)^q$ for $q$ even. Then,*

$$\Pr\left[\left|f(t) - \overline{f}\right| \geq \delta\right] \leq 2e \times \exp\left(-\frac{\delta}{eg}\right). \tag{8}$$

*Proof.* Let us set $\overline{f} = 0$ for simplicity, and focus on the case $x \geq \delta$. We have that

$$\Pr[\langle A(t)\rangle \geq \delta] = \int_{x \geq \delta} P(x)\mathrm{d}x \tag{9}$$

$$\leq \frac{1}{\delta^q} \int_{x \geq \delta} x^q P(x)\mathrm{d}x \tag{10}$$

$$\leq \frac{\mu_q}{\delta^q} \tag{11}$$

$$\leq \left(\frac{qg}{\delta}\right)^q . \tag{12}$$

A similar inequality holds for $x \leq -\delta$, so that $\Pr\left[\left|f(t)\right| \geq \delta\right] \leq 2\left(\frac{qg}{\delta}\right)^q$. The bound is obtained by choosing $q = \lfloor \frac{\delta}{eg} \rfloor$. $\qquad\square$

We now simply need to find the corresponding $g$ for the concentration bound to hold, which we do for various physical problems.

## 3 Generic models

First we consider models governed by a Hamiltonian $H = \sum_{m=1}^{D} E_m |E_m\rangle\langle E_m|$, which we assume to have a discrete and *generic* spectrum.

**Definition 1.** *Let $H$ be a Hamiltonian with spectrum $H = \sum_j E_j |E_j\rangle\langle E_j|$, and let $\Lambda_q, \Lambda_q'$ be two arbitrary sets of $q$ energy levels $\{E_j\}$. The spectrum of $H$ is generic if for all $q \in \mathbb{N}$ and all $\Lambda_q, \Lambda_q'$, the equality*

$$\sum_{j \in \Lambda_q} E_j = \sum_{j \in \Lambda_q'} E_j , \tag{13}$$

*implies that $\Lambda_q = \Lambda_q'$.*

This condition is expected to hold in non-integrable and chaotic models, such as those with Wigner-Dyson level statistics, and is otherwise expected to not hold in integrable models. It has previously appeared in the literature under various names [18,19]. It is an extension of the well-known non-degenerate gaps condition, which is the $q = 2$ case [1,20], and also implies that the energy spectrum is non-degenerate. A possible stronger condition, that implies Def. 1, is that of rational independence of the energy levels (see e.g. [21,22]). Notably, the probability of uniformly choosing a non-generic Hamiltonian is zero, as seen in the following lemma.

**Lemma 2.** *For any positive integer $d \geq 2$, the set of $d \times d$ complex Hermitian matrices that are not generic has Lebesgue measure zero.*

The proof is a straightforward generalization of the $q = 2$ case in [23] and can be found in Appendix B.

Consider $f(t) = \langle\psi|A(t)|\psi\rangle$ to be the pure state time evolution of some observable $A$. The first concentration result is as follows.

**Theorem 1.** *Let $H$ have a generic spectrum, with $\omega$ the diagonal ensemble, and $||A||$ the largest singular value of $A$. The moments in Eq. 4 are such that*

$$\mu_q \leq \left(q||A||\sqrt{\mathrm{Tr}[\omega^2]}\right)^q . \tag{14}$$

We thus have the bound

$$\Pr\left[\left|\langle A(t)\rangle - \overline{\langle A \rangle}\right| \geq \delta\right] \leq 2e \times \exp\left(-\frac{\delta}{e\|A\|\sqrt{\mathrm{Tr}[\omega^2]}}\right).$$

This states that the probability of finding $\langle A(t)\rangle$ away from $\overline{\langle A\rangle}$ even by a small amount is exponentially suppressed in $\sqrt{\mathrm{Tr}[\omega^2]}$. It also rigorously proves one of the main results from [3], where a similar concentration bound was heuristically argued as a consequence of the ETH. Previous results [13, 20] only yield the bound

$$\Pr\left[\left|\langle A(t)\rangle - \overline{\langle A \rangle}\right| \geq \delta\right] \leq \frac{\|A\|^2 \mathrm{Tr}[\omega^2]}{\delta^2}. \tag{15}$$

A particular observable of interest is the initial state itself, $A = |\Psi\rangle\langle\Psi|$. In this case, the quantity at hand is the fidelity with the initial state

$$F(t) = \langle\Psi|e^{-itH}|\Psi\rangle\langle\Psi|e^{itH}|\Psi\rangle. \tag{16}$$

**Theorem 2.** *Let $H$ have a generic spectrum and let $A = |\Psi\rangle\langle\Psi|$, then*

$$\mu_q \leq \left(q\,\mathrm{Tr}\left[\omega^2\right]\right)^q. \tag{17}$$

Notably, assuming a generic spectrum, the average fidelity is $\overline{F} = \mathrm{Tr}\left[\omega^2\right]$, so that we have the concentration bound

$$\Pr\left[\left|F(t) - \mathrm{Tr}[\omega^2]\right| \geq \delta\right] \leq 2e \times \exp\left(-\frac{\delta}{e\mathrm{Tr}[\omega^2]}\right). \tag{18}$$

This improves on Eq. (3) by a factor of $\sqrt{\mathrm{Tr}[\omega^2]}$ when substituted into $A = |\Psi\rangle\langle\Psi|$ and $\|A\| = 1$. Eq. (17) appeared previously in [22].

It is well known that $\mathrm{Tr}\left[\omega^2\right]$ is exponentially suppressed in system size for generic models for sufficiently well behaved initial conditions. As an example, see figure 2, which shows a clear exponential decay of $\mathrm{Tr}\left[\omega^2\right]$ with system size $L$.

## 4  Free fermions

The second class that we consider are extended free fermionic models

$$H = \sum_{m,n=1}^{L} M_{m,n} f_m^\dagger f_n, \tag{19}$$

where $f_n$ is a fermionic annihilation operator for the lattice site $n$. The fermionic operators obey the standard canonical anti-commutation relations $\{f_m, f_n\} = \{f_m^\dagger, f_n^\dagger\} = 0, \ \{f_m^\dagger, f_n\} = \delta_{m,n}$. We assume $M$ is real symmetric, so it is diagonalized with a real orthogonal matrix $O$ such that $M = ODO^T$. $D$ is a diagonal matrix with entries $D_{k,k} = \epsilon_k$, which allows us to rewrite the Hamiltonian as

$$H = \sum_{k=1}^{L} \epsilon_k d_k^\dagger d_k, \tag{20}$$

where $\epsilon_k$ are the single particle energy eigenmodes, and we have new fermionic operators in eigenmode space defined in terms of the real space fermionic operators: $d_k = \sum_{j=1}^{L} O_{j,k} f_j$. This class of models notably does not obey Def. 1. However, we can instead give the following definition.

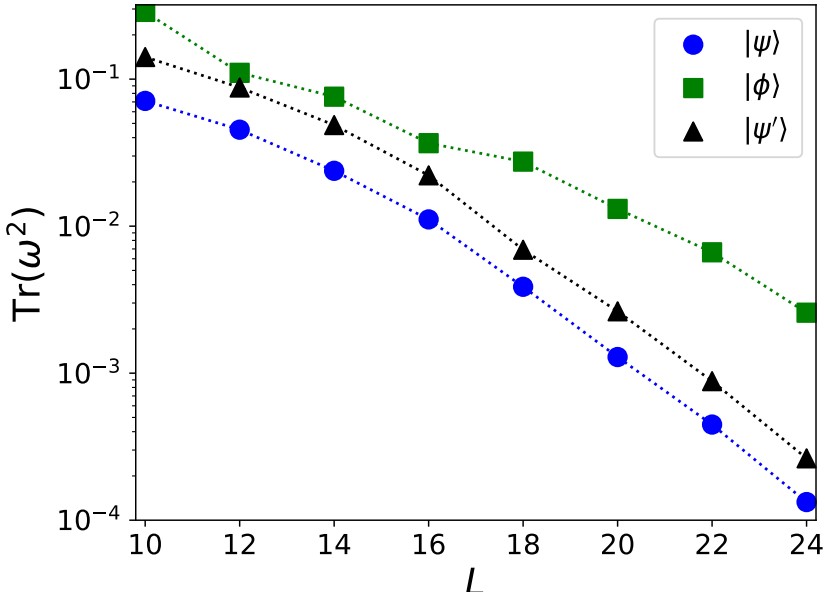

Figure 2: $\mathrm{Tr}[\omega^2]$ for a variety of system sizes and states. Numerics were done with the same model as Fig. 1 , which is non-integrable. The three states studied are $|\psi\rangle := |\uparrow\downarrow\uparrow\downarrow \ldots\ldots\rangle$, $|\psi'\rangle := \frac{1}{\sqrt{2}}(|\uparrow\downarrow\uparrow\downarrow \ldots\ldots\rangle + |\downarrow\uparrow\downarrow\uparrow \ldots\ldots\rangle)$ and $|\phi\rangle := \frac{1}{\sqrt{L}}\sum_{r=0}^{L-1}\hat{T}^r|\uparrow\uparrow \ldots \uparrow\downarrow \ldots \downarrow\downarrow\rangle$ where $\hat{T}$ is the translation operator shifting lattice indices by one. More details on the states and the model can be found in App. D.

**Definition 2.** *Let $H = \sum_{k=1}^{L} \epsilon_k d_k^\dagger d_k$ be a free Hamiltonian. Let $\Lambda_q, \Lambda_q'$ be two arbitrary sets of q eigenmodes $\{\epsilon_j\}$. Then H is an extended free fermionic model with a generic spectrum if, for all $q \in \mathbb{N}$ and all $\Lambda_q, \Lambda_q'$, the equality*

$$\sum_{j\in\Lambda_q} \epsilon_j = \sum_{j\in\Lambda_q'} \epsilon_j, \tag{21}$$

*implies that $\Lambda_q = \Lambda_q'$ and the entries of O are such that*

$$O_{j,k} = \frac{c_{j,k}}{\sqrt{L}}, \tag{22}$$

*with $c_{j,k} = \mathcal{O}(1)$.*

Generic free extended models can be constructed relatively easily by imposing translation invariance. Such models always have $O_{j,k} \propto 1/\sqrt{L}$. One can then construct an eigenmode spectrum $\epsilon_k$ that is generic. It would be interesting for future work to find local, or approximately local examples of such a model.

The equivalent of Lemma 2 also holds here by applying it to the matrix $M$ and the energy eigenmodes. This definition crucially excludes localized models, which have entries of the form, $O_{m,k} \sim e^{-|k-m|/\xi}$, with $\xi$ the localization length. The bound on the moments is as follows.

**Theorem 3.** *Let H be an extended free fermionic Hamiltonian with generic spectrum and let $A = f_m^\dagger f_n$. Then, for even q,*

$$\kappa_q \leq \left(qc^2\sqrt{\frac{\nu}{L}}\right)^q, \tag{23}$$

*where $\nu = \frac{N}{L}$ is the filling factor of the fermions on the lattice and $c = \sqrt{L}\max_{k_j}\{O_{m,k_j}, O_{n,k_j}\}$.*

The corresponding concentration bound is

$$\Pr\left[\left|\langle f_m^\dagger f_n(t)\rangle - \overline{\langle f_m^\dagger f_n\rangle}\right| \geq \delta\right] \leq 2e \times \exp\left(-\frac{\delta}{ec^2}\sqrt{\frac{L}{\nu}}\right).$$

Theorem 3 can be contrasted with the bound found in [24] for the second moment. The authors consider a potentially extensive observable and do not limit the analysis to extensive models, recovering $\kappa_2 \leq \|a\|^2 \nu L$ for an observable $A = \sum_{m,n} f_m^\dagger a_{m,n} f_n$.

The last quantity of interest is the single particle propagator

$$\{f_m^\dagger(t), f_n\} = a_{m,n}(t).$$

For example, if we initialize our state as $|\Psi\rangle = f_m^\dagger|0\rangle$, then the fidelity is

$$F(t) = |a_{m,m}(t)|^2. \tag{24}$$

The more general $|a_{m,n}(t)|^2$ is also studied in the context of out of time ordered correlators [25–28]. Consider

$$|a_{m,n}(t)|^2 = \sum_{k,l} O_{m,k} O_{n,k} O_{m,l} O_{n,l} e^{i(\epsilon_k - \epsilon_l)t}. \tag{25}$$

The infinite time average of this quantity is taken as

$$\omega_{m,n} = \sum_k O_{m,k}^2 O_{n,k}^2. \tag{26}$$

For extended models with non-degenerate frequencies this quantity decays to zero since $\omega_{m,n} \sim \frac{1}{L}$ and $\kappa_2 \leq \frac{c}{L^2}$, where $c$ is weakly dependent on system size and is $\mathcal{O}(1)$ in the thermodynamic limit [26]. We can bound the moments for the single particle propagator as follows.

**Theorem 4.** *Let $H$ be a free fermionic Hamiltonian with a generic spectrum, and let the dynamical function $f(t) = |a_{m,n}(t)|^2$ be the squared single particle propagator. The moments are then bounded by*

$$\kappa_q \leq \left(\frac{qc^4}{L}\right)^q, \tag{27}$$

*where $c = \sqrt{L} \max_{k_j}\{O_{m,k_j}, O_{n,k_j}\}$.*

Finally, the corresponding concentration bound is

$$\Pr\left[\left||a_{m,n}(t)|^2 - \omega_{m,n}\right| \geq \delta\right] \leq 2e \times \exp\left(-\frac{\delta L}{ec^4}\right). \tag{28}$$

## 5 Recurrence time

All the quantities analyzed above always come back arbitrarily close to their initial values at $t = 0$. For large systems, however, such recurrences only happen at astronomically large timescales, inaccessible to both experiments and numerical studies. We now put a lower bound on those timescales through a suitably defined notion of *average recurrence time*, both for observables and also the whole state.

**Definition 3.** *A $(u, \Delta, A)$-recurrence occurs at a time interval $\mathcal{C}_\Delta = [t_\Delta, t_\Delta + \Delta]$ if, for all $t \in \mathcal{C}_\Delta$,*

$$|\langle A(t)\rangle - \langle A(0)\rangle| \leq u\|A\|, \tag{29}$$

*where $\Delta > 0$ and $u \in (0, 2]$. Similarly, a $(u, \Delta)$-recurrence occurs if, for all $t \in \mathcal{C}_\Delta$,*

$$1 - F(t) \leq u. \tag{30}$$

We identify a recurrence with an interval of time during which the system is close to the initial value. Thus, the definition contains two free parameters: $\Delta$ is the time duration of that recurrence, and $u$ quantifies how close to the initial state that is. We expect that the larger $\Delta$ and the smaller $u$ are, the more seldom a $(u, \Delta, A)$-recurrence will happen. In fact, for some large enough $\Delta$, a $(u, \Delta, A)$-recurrence will likely never happen.

Notice that it follows from the Fuchs-van de Graafs inequalities [29] that an $(u, \Delta)$-recurrence implies an $(u, \Delta, A)$-recurrence for all $A$, and that conversely an $(u, \Delta, A)$-recurrence for all $A$ implies an $(u, \Delta)$-recurrence. However, individual observables may have additional earlier recurrences than those of the fidelity.

Let us also define $t_\Delta^n(A)$ as the initial point of the time interval corresponding to the $n$-th $(u, \Delta, A)$-recurrence, so that $t_\Delta^n(A) < t_\Delta^{n+1}(A)$, and analogously, $t_\Delta^n$ for the fidelity recurrences. This motivates the following definition, inspired by that in [10].

**Definition 4.** *The average $(u, \Delta, A)$-recurrence time is*

$$T(u, \Delta, A) \equiv \lim_{n \to \infty} \frac{t_\Delta^n(A)}{n}, \tag{31}$$

*with $T(u, \Delta)$ analogously defined.*

$T(u, \Delta, A)$ can be understood as the inverse of the density of recurrences. To see this, first note that $\langle A(t) \rangle$ is a Besicovitch almost periodic function and we can therefore choose a well-defined almost period $\tilde{P}$ in the sense of [30]. Evaluating our sequence at integer multiples of the almost period gives $\frac{t_\Delta^n(A)}{n} \to \frac{\tilde{P}}{n_{\tilde{P}}}$ where $n_{\tilde{P}}$ is the number of $t_\Delta^n(A)$ inside an almost period. Then it follows that $T(u, \Delta, A) = 1/D(u, \Delta, A)$ where $D(u, \Delta, A)$ is the density of $(u, \Delta, A)$ recurrences. These quantities can be easily bounded with the concentration bounds above. First, for $T(u, \Delta, A)$.

**Corollary 1.** *Let $H$ have a generic spectrum, and let w.l.o.g. $\langle A(0) \rangle - \overline{\langle A \rangle} = c_A ||A|| \geq 0$. Then, for $u \leq c_A$,*

$$\frac{\Delta}{2e} \exp\left(\frac{c_A - u}{e\sqrt{\mathrm{Tr}[\omega^2]}}\right) \leq T(u, \Delta, A). \tag{32}$$

*Proof.* From the definition of the distribution $P(x)$ in Eq. (6) and Eq. (3) we have that

$$\lim_{n \to \infty} \frac{\Delta n}{t_\Delta^n(A) + \Delta} \leq \Pr\left[|\langle A(t) \rangle - \langle A(0) \rangle| \leq u ||A||\right]$$

$$\leq \Pr\left[|\langle A(t) \rangle - \overline{\langle A \rangle}| \geq (c_A - u) ||A||\right]$$

$$\leq 2e \times \exp\left(\frac{u - c_A}{e\sqrt{\mathrm{Tr}[\omega^2]}}\right). \tag{33}$$

Notice that $\lim_{n \to \infty} \frac{\Delta n}{t_\Delta^n(A) + \Delta} = \Delta / T(u, \Delta, A)$. Solving for $T(u, \Delta, A)$ yields the result. $\qquad \square$

The previous results are stated in terms of arbitrary recurrences, defined with free parameters $\Delta, u$. We now explain for which choices of these we expect the bounds to be more meaningful. With typical out-of-equilibrium initial conditions, we have that $c_A = \mathcal{O}(1)$. In that case, meaningful recurrences, comparable in magnitude to the initial condition, will be such that $c_A - u = \mathcal{O}(1)$. We expect that in those cases, the recurrences will have a duration comparable to that of the initial equilibration time $T_{\mathrm{eq}}^A$, which we can loosely define as the time it takes for $\langle A(t) \rangle$ to initially settle around the steady value $\overline{\langle A \rangle}$. These recurrences are then on average spaced by a time which we roughly estimate to be

$$T \gtrsim T_{\mathrm{eq}}^A e^{\Omega\left(\mathrm{Tr}[\omega^2]^{-1/2}\right)}. \tag{34}$$

See [31–35] for analytical bounds on $T_{eq}^A$. Note that for local Hamiltonians and observables, $T_{eq}^A$ is believed to generally scale as a low-degree polynomial in system size [36].

A possible shortcoming of Eq. (32) is that the bound vanishes when $\Delta \to 0$. That is, Corollary 1 is unable to estimate vanishingly small recurrences. However, we expect that meaningful recurrences, getting somewhat close to the expectation value of initial conditions, will typically have a relaxation time comparable to that of the initial conditions, in which case the estimate of Eq. (34) applies.

For the fidelity, the bound on the recurrence follows exactly the proof of Corollary 1 but using Eq. (18) instead.

**Corollary 2.** *Let H have a generic spectrum. Then,*

$$\frac{\Delta}{2e^2}\exp\left(\frac{1-u}{e\,\mathrm{Tr}[\omega^2]}\right) \le T(u,\Delta).\tag{35}$$

Again, there are particular choices of the free parameters for which we expect the bounds to be more meaningful. In many-body systems, the fidelity initially decays as $F(t) = e^{-\sigma^2 t^2/2}$ where $\sigma^2 = \langle\Psi|H^2|\Psi\rangle - \langle\Psi|H|\Psi\rangle^2$ is the energy variance [37–39]. Recurrences with $u = \mathcal{O}(1)$, in which the time-evolved state reaches a meaningful fidelity with the initial one, likely decay in a similar fashion, and as such we expect them to have a time duration of $\Delta \sim \sigma^{-1}$, so that on average they are spaced by a time

$$T \gtrsim \sigma^{-1} e^{\Omega\left(\mathrm{Tr}[\omega^2]^{-1}\right)}.\tag{36}$$

This resembles the result in [10], which gives an exact calculation of the average recurrence time based on some heuristic assumptions on the wave-function, and finds a similar scaling of $e^{\Omega\left(\mathrm{Tr}[\omega^2]^{-1}\right)}$, with a slightly different prefactor. It also matches the scaling of other previous estimates [9], so Eq. (35) should be close to optimal.

Finally, we also have corresponding bounds for fermions.

**Corollary 3.** *Let H be a free fermionic Hamiltonian with generic spectrum, and let w.l.o.g.* $\langle f_m^\dagger f_n(0)\rangle - \overline{\langle f_m^\dagger f_n\rangle} = c_f \ge 0$. *Then, for* $u \le c_f$,

$$\frac{\Delta}{2e}\exp\left(\frac{c_f-u}{ec^2}\sqrt{\frac{L}{v}}\right) \le T(u,\Delta,f_m^\dagger f_n),\tag{37}$$

as well as for the fidelity in Eq. (24).

**Corollary 4.** *Let H be a free fermionic Hamiltonian with generic spectrum. Then,*

$$\frac{\Delta}{2e}\exp\left(\frac{(1-u)L}{ec^4}\right) \le T(u,\Delta).\tag{38}$$

The analogue of Eq. (34) and Eq. (36) also holds following the same considerations. These bounds however scale as $e^{\Omega(\sqrt{L})}$ and $e^{\Omega(L)}$ respectively, which are exponential in the number of sites $L$. This is a fast scaling, but still exponentially slower than that from Corollaries 1 and 2. Even shorter recurrence times are also found in specific instances of Bose gases [40–42], which can even be experimentally tested [43] with cold atoms.

# 6  Conclusion

We have shown how in systems with a generic spectrum as given in Definition 1 both observables and the fidelity with the initial state equilibrate around their time-averaged values, with out-of-equilibrium fluctuations suppressed exponentially in the effective dimension $\mathrm{Tr}\left[\omega^2\right]^{-1}$. This number scales exponentially under very general conditions on the state and the Hamiltonian [23, 44–46], so in these systems fluctuations are most often doubly exponentially suppressed. Since partial or full recurrences are far from equilibrium fluctuations, our bounds yield an estimate of their occurrence, with a scaling that we believe is almost optimal. Equivalent results with a slower scaling also hold for free fermions.

Previous works [23, 44, 46–49] start with the bound on the second moment in [13, 14] to obtain results on equilibration, so the present findings naturally strengthen them. Also, Theorem 2 in [50] extends [13, 14] to two-point correlation functions, and the corresponding concentration bound is straightforward.

Our bounds on the recurrence time apply to individual states. A given Hamiltonian should also have other later state-independent recurrences. For instance, the recent result for random circuits [51] suggests that a recurrence in complexity of $e^{-itH}$ might still doubly exponential, but with a larger exponent that Eq. (36).

# Acknowledgments

**Funding information**  AMA acknowledges support from the Alexander von Humboldt foundation. J.R. and N.J.P. acknowledges support from the Natural Sciences and Engineering Research Council of Canada (NSERC).

# A  Defining moments

The average value of $\langle A(t)\rangle$ in Eq. 1 motivates the formal definition of the following probability distribution

$$P(x) = \lim_{T\to\infty}\int_0^T \frac{\mathrm{d}t}{T}\delta(x - \langle A(t)\rangle)\,.$$

Note that this integral is convergent since this distribution is well-defined by the argument given between equations 4-6. We would like to compute the moments as

$$\mu_q \equiv \lim_{T\to\infty}\int_0^T \frac{\mathrm{d}t}{T}\left(\langle A(t)\rangle - \overline{\langle A\rangle}\right)^q\,.$$

This requires swapping the integral over $x$ and limit in $T$ and can be justified using the dominated convergence theorem. To use this famous result we must prove that the absolute value of the integrand is bounded by an integrable function. Consider

$$\kappa_q = \int x^q \lim_{T\to\infty} P_T(x)dx\,.$$

For book-keeping purposes let $g(t) = \left(\langle A(t)\rangle - \overline{\langle A\rangle}\right)$. We may bound the integrand inside the limit

$$|x^q P_T(x)| = |\frac{x^q}{T}\int_0^T \mathrm{d}t\,\delta(x - g(t))| \leq \frac{|x|^q}{T}\int_0^T \mathrm{d}t\,\delta(x - g(t))\,.$$

Integrating the absolute value of the right-hand side and applying Fubini's theorem we find

$$\int \left| \frac{|x|^q}{T} \int_0^T \mathrm{d}t\, \delta(x-g(t)) \right| dx \le \int \frac{|x|^q}{T} \left| \int_0^T \mathrm{d}t\, \delta(x-g(t)) \right| dx$$

$$\le \int \frac{|x|^q}{T} \int_0^T \mathrm{d}t\, \delta(x-g(t)) dx$$

$$\le \frac{1}{T} \int_0^T \mathrm{d}t\, |g(t)|^q \,.$$

The function $g(t)$ is continuous and therefore bounded on the interval $[0, T]$, so the integral is finite, therefore its positive and negative components are as well, thus the $q$-th absolute moment is bounded. We may therefore conclude that the absolute value of the integrand is bounded by an integrable function, so by the dominated convergence theorem we may swap the integral and limit.

Note that equivalently, one may define the moments for finite $T$ and then take the limit.

# B  Generic spectra

**Lemma 1.** *The set of generic Hermitian matrices in $M_d(\mathbb{C})$ has full Lebesgue measure.*

*Proof.* Note that this proof is similar to the proof that the set of non-diagonalizable matrices has Lebesgue measure zero. Let $H$ be some Hermitian $d \times d$ matrix. We start by defining the function

$$F(H) = \prod_{\substack{n_1,m_1,\dots,n_q,m_q \\ n_i \ne m_i}} \left( \sum_{i=1}^q E_{n_i} - E_{m_i} \right).$$

This function is zero precisely when the spectrum is not generic. Clearly swapping eigenvalues does not change the function $F$, i.e. $F$ is a symmetric polynomial of the eigenvalues. By the fundamental theorem of symmetric (real) polynomials $F$ can be written uniquely as a polynomial in the elementary symmetric polynomials in $E_i$'s, which are precisely trace powers of $H$.

Recall that $H$ can be expressed in some basis. For example some generalized Pauli basis, or even the standard basis. This is conceptually equivalent to saying that the vector space of all possible $H$ can be parameterized by the coefficients of the basis elements in the expansion. Thus, we may expand $H$ in this basis and then take trace powers, showing us that $F$ is a real polynomial in the space of these coefficients.

It is a well known fact from measure theory that the zero set of a multivariate polynomial has Lebesgue measure zero. □

# C  Bounding the moments

## C.1  Proof for generic models

**Theorem 1.** *Let $H$ have a generic spectrum, $\omega$ the diagonal ensemble, and $||A||$ the largest singular value of A. The moments in Eq. 4 in the main text are such that*

$$\mu_q \le \left( q||A|| \sqrt{\mathrm{Tr}[\omega^2]} \right)^q. \tag{C.1}$$

*Proof.* First we prove the following inequality:

$$|\text{Tr}[(A\omega)^q]| \leq \left(||A||\sqrt{\text{Tr}[\omega^2]}\right)^q. \tag{C.2}$$

To realize this, consider the matrix $A\omega$, which is not Hermitian and therefore may have complex eigenvalues and may potentially not be diagonalizable. This, however, does not prevent us from finding a complete set of eigenvalues such that their multiplicity summed is the dimension of $A\omega$. The matrix is always similar to its Jordan form, and we can in general always write

$$\text{Tr}[A\omega] = \sum_i \lambda_i, \tag{C.3}$$

where $\lambda_i$ is the $i$-th (potentially complex) eigenvalue of $A\omega$. More generally, we can always write

$$\text{Tr}[(A\omega)^q] = \sum_i \lambda_i^q. \tag{C.4}$$

From here we can bound the following:

$$|\text{Tr}[(A\omega)^q]| = |\sum_i \lambda_i^q| \tag{C.5}$$

$$\leq \sum_i |\lambda_i^q| \tag{C.6}$$

$$= ||\lambda||_q^q \tag{C.7}$$

$$\leq ||\lambda||_2^q = \left(\sum_i |\lambda_i|^2\right)^{q/2}, \tag{C.8}$$

where we used the triangle inequality in (C.6), and in (C.8) we use the property $||x||_{p+a} \leq ||x||_p$ for any vector $x$ and real numbers $p \geq 1$ and $a \geq 0$. Note that, using the Shur decomposition, we may write $A\omega = QUQ^\dagger$, where $Q$ is a unitary matrix, and $U$ is upper triangular with the same spectrum on the diagonal as $A\omega$. Using this, we have that

$$\text{Tr}\left[(A\omega)(A\omega)^\dagger\right] = \text{Tr}\left[QUQ^\dagger(QUQ^\dagger)^\dagger\right] = \text{Tr}\left[UU^\dagger\right] = \sum_i |\lambda_i|^2 + \text{other non-negative terms.}$$

Thus,

$$\sum_i |\lambda_i|^2 \leq \text{Tr}\left[A\omega\omega^\dagger A^\dagger\right] \leq ||AA^\dagger||\,\text{Tr}\left[\omega\omega^\dagger\right] \leq ||A||^2\,\text{Tr}\left[\omega\omega^\dagger\right] = ||A||^2\,\text{Tr}\left[\omega^2\right].$$

This gives us our desired inequality.

Moving on, let us derive a general bound for the $q$-th moment of models satisfying Definition 1. For simplicity, and w.l.o.g., let us assume that $\overline{\langle A \rangle} = 0$. Expanding the definition of the moments we arrive at

$$\mu_q = \lim_{\tau \to \infty} \frac{1}{\tau} \int_0^\tau dt \sum_{m_1, n_1, \dots, m_q, n_q} \prod_{i=1}^q \left(A_{m_i, n_i} \bar{c}_{m_i} c_{n_i}\right) e^{i(E_{m_i} - E_{n_i})t}. \tag{C.9}$$

The assumption that the Hamiltonian is generic means that only certain terms in the sum survive after averaging over all time: those for which the sets of $\{m_i\}$ and $\{n_i\}$ coincide up to permutations, which we denote with $\{\sigma(i)\}$. Due to the equilibrium expectation value being zero, we can also eliminate all terms for which $i = \sigma(i)$ for $1 \leq i \leq q$. Thus, we want all permutations on $q$ elements except those that have a fixed point. Such permutations are

called derangements. The number of distinct derangements is denoted by $!q = \lfloor \frac{q!}{e} + \frac{1}{2} \rfloor$. Let $D_q$ denote the set of derangements on $\{1, 2, .., q\}$. Eq. (C.9) becomes

$$\mu_q = \sum_{m_1,...,m_q} \prod_{i=1}^{q} |c_{m_i}|^2 \sum_{\sigma \in D_q} \prod_{i=1}^{q} A_{m_i, \sigma(m_i)}. \tag{C.10}$$

Given a derangement $\sigma$, it can be decomposed as the product of cycles $\sigma_1, \sigma_2, ..., \sigma_r$ with lengths $\ell_1, \ell_2, ..., \ell_r$, respectively, such that $\sum_{j=1}^{r} l_j = q$. In each term of the inner summation we can collect terms of the same cycle. For example for $q = 6$ and $\sigma = \sigma_1 \sigma_2 = (m_1, m_2)(m_3, m_4, m_5, m_6)$ the term can be written as

$$(A_{m_1, m_2} A_{m_2, m_1})(A_{m_3, m_4} A_{m_4, m_5} A_{m_5, m_6} A_{m_6, m_3}).$$

Summing over $m_1, m_2, m_3, m_4, m_5, m_6$ we have that this term is precisely

$$\mathrm{Tr}\left[(A\omega)^2\right] \mathrm{Tr}\left[(A\omega)^4\right].$$

In general, each cycle of a term will correspond to a product of trace powers i.e. $\sigma = \sigma_1, \sigma_2, ..., \sigma_r$ corresponds to

$$\mathrm{Tr}\left[(A\omega)^{\ell_1}\right] \mathrm{Tr}\left[(A\omega)^{\ell_2}\right] ... \mathrm{Tr}\left[(A\omega)^{\ell_r}\right].$$

We may apply Eq. (C.2) term-wise to get

$$\mathrm{Tr}\left[(A\omega)^{\ell_1}\right] \mathrm{Tr}\left[(A\omega)^{\ell_2}\right] ... \mathrm{Tr}\left[(A\omega)^{\ell_r}\right] \leq \left(||A|| \sqrt{\mathrm{Tr}[\omega^2]}\right)^{\ell_1 + \ell_2 + ... + \ell_r} = \left(||A|| \sqrt{\mathrm{Tr}[\omega^2]}\right)^{q}.$$

In each moment's inner summation there are precisely $!q$ terms of this form because there are $!q$ derangements, thus

$$\mu_q \leq !q \left(||A|| \sqrt{\mathrm{Tr}[\omega^2]}\right)^{q} \leq \left(q||A|| \sqrt{\mathrm{Tr}[\omega^2]}\right)^{q}. \tag{C.11}$$

$\square$

The $q = 2$ case can be found in [20].

**Theorem 2.** *Let $H$ have a generic spectrum and let $A = |\Psi\rangle \langle\Psi|$, then*

$$\mu_q \leq \left(q \, \mathrm{Tr}[\omega^2]\right)^{q}. \tag{C.12}$$

*Proof.* The moments defined for the fidelity are defined as

$$\mu_q = \lim_{\tau \to \infty} \frac{1}{\tau} \int_0^{\tau} dt \prod_{i=1}^{q} \sum_{m_i \neq n_i} |c_{m_i}|^2 |c_{n_i}|^2 e^{i(E_{m_i} - E_{n_i})t}, \tag{C.13}$$

$$= \sum_{m_1,...m_q} \prod_{i=1}^{q} |c_{m_i}|^2 \prod_{\sigma \in D_q} |c_{\sigma(m_i)}|^2. \tag{C.14}$$

In the above expression we can note that there are $!q$ possible derangements using genericity given in Definition 1. Each $m_i$ will have one pair given to us from $\sigma(m_i)$, implying each individual term in the sum is $\mathrm{Tr}[\omega^2]^q$, so our final expression is

$$\mu_q = !q \, \mathrm{Tr}[\omega^2]^q \leq \left(q \, \mathrm{Tr}[\omega^2]\right)^{q}. \tag{C.15}$$

$\square$

## C.2 Generic free models

This class of models conserves total particle number, which we will denote as

$$N = \sum_{j=1}^{L} \langle f_j^\dagger f_j \rangle = \sum_{k=1}^{L} \langle d_k^\dagger d_k \rangle. \tag{C.16}$$

**Theorem 3.** *Let $H$ be a extended free fermionic Hamiltonian with a generic spectrum and let $A = f_m^\dagger f_n$. Then, the even moments are bounded above by*

$$\kappa_q \le \left( qc^2 \sqrt{\frac{\nu}{L}} \right)^q, \tag{C.17}$$

*where $\nu = \frac{N}{L}$ is the filling factor of the fermions on the lattice and $c = \sqrt{L} \max_{k_j} \{ O_{m,k_j}, O_{n,k_j} \}$.*

*Proof.* Consider

$$\kappa_{2n} = \lim_{T \to \infty} \frac{1}{T} \int_0^\infty dt \prod_{j=1}^{n} \sum_{k_j \ne l_j} O_{m,k_j} O_{n,l_j} \langle d_{k_j}^\dagger d_{l_j} \rangle e^{i(\epsilon_{k_j} - \epsilon_{l_j})t} \sum_{p_j \ne q_j} O_{m,p_j} O_{n,q_j} \langle d_{q_j}^\dagger d_{p_j} \rangle e^{i(\epsilon_{q_j} - \epsilon_{p_j})t}. \tag{C.18}$$

Let us define the tensor (note the $j$ dependence)

$$B_{k_j,l_j} = \begin{cases} O_{m,k_j} O_{n,l_j} \langle d_{k_j}^\dagger d_{l_j} \rangle, & j \text{ odd}, \\ O_{m,l_j} O_{n,k_j} \langle d_{k_j}^\dagger d_{l_j} \rangle, & j \text{ even}, \\ 0, & k_j = l_j. \end{cases} \tag{C.19}$$

This allows us to rewrite our equation as

$$\kappa_{2n} = \lim_{T \to \infty} \frac{1}{T} \int_0^\infty dt \prod_{j=1}^{2n} \sum_{k_j,l_j} B_{k_j,l_j} e^{i(\epsilon_{k_j} - \epsilon_{l_j})t}. \tag{C.20}$$

This can likewise be rewritten as

$$\kappa_{2n} = \lim_{T \to \infty} \frac{1}{T} \int_0^\infty dt \sum_{k_1,l_1,\dots k_{2n},l_{2n}} \prod_{j=1}^{2n} B_{k_j,l_j} e^{i(\epsilon_{k_j} - \epsilon_{l_j})t}. \tag{C.21}$$

Assuming a generic single-particle spectrum, this means we have the following surviving terms:

$$\kappa_{2n} = \sum_{k_1,\dots k_{2n}} \sum_{\sigma \in S_{2n}} \prod_{j=1}^{2n} B_{k_j,\sigma(k_j)}, \tag{C.22}$$

where $S_{2n}$ denotes the symmetric group on $1, 2 \dots 2n$. We can then enforce the fact that these terms are zero if $k_j = \sigma(k_j)$ for $1 \le j \le 2n$. So denoting the derangements as $D_{2n}$ as earlier, we arrive at

$$\kappa_{2n} = \sum_{k_1,\dots k_{2n}} \sum_{\sigma \in D_{2n}} \prod_{j=1}^{2n} B_{k_j,\sigma(k_j)}. \tag{C.23}$$

Next, recognizing that each definition of $B$ contains two extensive terms multiplied, let $c = \sqrt{L} \max_{k_j} \{ O_{m,k_j}, O_{n,k_j} \}$,

$$\kappa_{2n} \le \frac{c^{4n}}{L^{2n}} \sum_{k_1,\dots k_{2n}} \sum_{\sigma \in D_{2n}} \prod_{j=1}^{2n} \langle d_{k_j}^\dagger d_{\sigma(k_j)} \rangle. \tag{C.24}$$

As in Theorem 1, each term will be a trace of powers of $\Lambda$, and there can be at most $n$ products of traces of $\Lambda$. Since $0 \leq \Lambda \leq \mathbb{I}$, each trace of $\Lambda$ can further be bounded by $\mathrm{Tr}[\Lambda^p] \leq \mathrm{Tr}[\Lambda] = N = \nu L$, which means we can bound $\kappa_{2n}$ by

$$\kappa_{2n} \leq !(2n)\frac{c^{4n}\nu^n}{L^{2n-n}} \leq \left(\frac{4n^2 c^4 \nu}{L}\right)^n, \tag{C.25}$$

where $c$ is weakly dependent on system size and $0 \leq \nu \leq 1$. Choosing $q = 2n$ and reorganizing gives the desired result. $\qquad\square$

**Theorem 4.** *Let H be a free fermionic Hamiltonian with a generic spectrum, and let our dynamical function $f(t) = |a_{m,n}(t)|^2$ be the squared single particle propagator, then we can bound the moments by*

$$\kappa_q \leq \left(\frac{qc^4}{L}\right)^q, \tag{C.26}$$

*where $c = \sqrt{L}\max_{k_j}\{O_{m,k_j}, O_{n,k_j}\}$.*

*Proof.* The $q$-th moment can be written as

$$\mu_q = \lim_{T \to \infty}\frac{1}{T}\int_0^\infty dt \prod_{i=1}^{q}\sum_{k_i \neq l_i} O_{m,k_i}O_{n,k_i}O_{m,l_i}O_{n,l_i}e^{i(\epsilon_{k_i}-\epsilon_{l_i})t}, \tag{C.27}$$

through the usual procedure and using the definition 2 in the main text we recover

$$\mu_q = \sum_{k_1,\ldots k_q}\prod_{i=1}^{q}O_{m,k_i}O_{n,k_i}\prod_{\sigma \in D_q}O_{m,\sigma(k_i)}O_{n,\sigma(k_i)}, \tag{C.28}$$

defining $c = \sqrt{L}\max_{k_j}\{O_{m,k_j}, O_{n,k_j}\}$, we factor out four of these, and sum up the indices, giving us

$$\mu_q \leq \frac{!qc^{4q}}{L^q} \leq \left(\frac{qc^4}{L}\right)^q. \tag{C.29}$$

$\qquad\square$

# D  Numerics

The numerics for the figures in the main body were carried out on the spin 1/2 Hamiltonian,

$$H = \sum_{j=1}^{L}J_1\left(S_j^+ S_{j+1}^- + \mathrm{h.c}\right) + \gamma_1 S_j^Z S_{j+1}^Z + J_2\left(S_j^+ S_{j+2}^- + \mathrm{h.c}\right) + \gamma_2 S_j^Z S_{j+2}^Z,$$

where $(J_1, \gamma_1, J_2, \gamma_2) = (-1, 1, -0.2, 0.5)$ giving us a non-integrable model. We perform exact diagonalization exploiting total spin conservation and translation invariance. We choose pure initial states that allow us to further exploit the $Z_2$ spin flip symmetry and the spatial reflection symmetry. In Fig. 1 we see the approximated probability distribution function $\tilde{P}_T(x)$ as a histogram. The observable is $A = \sigma_1^Z$, the Pauli-z matrix on the first lattice site. The initial state is a Néel type state:

$$|\psi\rangle = |\uparrow\downarrow\ldots\rangle. \tag{D.1}$$

In Fig. 2 we calculate the purity of the diagonal ensemble $\mathrm{Tr}\left[\omega^2\right]$ for three states. The states featured are

$$|\psi\rangle := |\uparrow\downarrow\uparrow\downarrow\ldots\rangle, \tag{D.2}$$

$$|\psi'\rangle := \frac{1}{\sqrt{2}}\left(|\uparrow\downarrow\uparrow\downarrow\ldots\rangle + |\downarrow\uparrow\downarrow\uparrow\ldots\rangle\right), \tag{D.3}$$

$$|\phi\rangle := \frac{1}{\sqrt{L}}\sum_{r=0}^{L-1}\hat{T}^r|\uparrow\uparrow\ldots\uparrow\downarrow\ldots\downarrow\downarrow\rangle. \tag{D.4}$$

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
