# Peer review of "Concentration of quantum equilibration and an estimate of the recurrence time"

_SciPost Physics, doi:SciPost Phys. 15, 165 (2023)_

## Round 1 · Referee Report · Lorenzo Campos Venuti (Referee 1) · 2023-3-10

Strengths

1- Substantially improved previous result 2- Well written 3- Some techniques or arguments may be used in future works

Weaknesses

1- The definition of recurrence time depends on an apparently free parameter \Delta 2- Some proofs or arguments may be clarified a little more

Report

This paper is a solid work that considerably strengthen previous concentration results on the equilibration of isolated quantum systems. The main question in this field is the following: a quantum system is let evolve freely (i.e., unitarily) starting from an initial state (a pure state in the manuscript). One then monitors the expectation value of an observable A over time, <A(t)>. The question is how much time does <A(t)> spends near to its average value. Previous results had control over the variance of the related probability distribution while in the present work the authors are able to bound all the moments and the resulting concentration result is essentially exponentially better than previous estimates (Ref. [13]). Using this result the authors are able to find a lower bound to the Poincare recurrence time. The same results are also presented for quasi-free fermions with particle number conserving Hamiltonian.

In summary the results are interesting and the paper is well written.
I would like, however, that the authors addressed some comments prior to publication. In particular regarding the section "Recurrence time".

Requested changes

1) Eq. (1). The notation <A(\infty)> to indicate the infinite time average is not consistent with <A(t)>, as it is not the infinite time limit of the latter (the infinite time limit does not exist in the finite dimensional case considered). My suggestion, unless the authors have a strong case against, is to use a different notation. A common one is to use \overline{} to denote the infinite time average.

2) Page one, second column, third paragraph, "This gives, for the first time, a mathematically rigorous scaling on the average recurrence time, that matches the scaling of previous estimates [9, 10]. See [11, 12] for other results.". I am not super happy about the wording of this sentence. Some journals even discourage the use of "for the first time", "novel" and so on. In any case it is true that Refs. [9,12] provide estimates of the recurrence time, but Ref [10] gives an exact result for the recurrence time obtained using the fidelity [Eq. (17)]. Finally Ref [11] considers a time dependent, periodic, Hamiltonian, and so it is not relevant here (though it is no harm to cite it of course).

3) Eq. (3) has the same notational problem as Eq. (1) [see 1)]

4) Eq. (4). The use of \kappa_q to indicate centered moments is also a bit cumbersome and possibly misleading as the same letter is usually used to indicate cumulants. The standard notation for centered (or central) moments is \mu_q.

5) Eq. (5) a small typo. k_q is used in placed of \kappa_q

I have some concerns regarding the discussion in the section "Recurrence time".

6) In definition 3 I am not sure I understand the role of \Delta. As it is written \Delta is a free parameter, but it is clear that for large enough \Delta there would be no recurrence at all. Also the notation t_\Delta is confusing as the recurrence time does not seem to depend on \Delta. In short, I would like the authors to clarify the definition of recurrence. This perplexity continues in Definition 4 and in the results Eqns. (35) and (38): why the recurrence time depends on this free parameter ∆? In some instances it seems that ∆ plays a role similar to t^{n+1} - t^n. See also points 8) and 10) below.

7) Page 5 first column: "Notice that an (u, ∆)-recurrence implies an (u, ∆, A)- recurrence for all A, and that conversely an (u, ∆, A)- recurrence for all A implies an (u, ∆)-recurrence. However, individual observables may have additional earlier recurrences.". Can the authors please provide a proof a this assertion, perhaps in the appendix?

8) Following paragraph: "Let us also define t^n_∆(A) as the time for the n-th (u, ∆, A)-recurrence, so that t^n (A) < t^{n+1} (A), and analogously, t^n_∆ for the fidelity. ". This definition is not sufficient to define the countable set t^n_∆(A) (for all n=1,2,...). Even looking at definition 3 does not help as there are uncountable times t satisfying Eq. (32). Perhaps the authors meant to define t^n(A) (I'm dropping the subscript ∆) as the n-th solution of

|<A(t)>-<A(0)>| = u ||A||

(i.e., Eq. (32) with equality), in the interval t\in[0,\infty] ordered in increasing order.

9) Few lines below: "T(u, ∆, A) can be understood as the inverse of the density of recurrences. To see this, first note that ⟨A(t)⟩ is an almost periodic function and has an almost period P ̃. Evaluating our sequence at integer multiples of the almost period gives t^n_∆ (A)/n → P ̃/n_P where n_P is the number of t^n_∆(A) inside an almost period. Then it follows that T(u,∆,A) = 1/D(u,∆,A) where D(u,∆,A) is the density of (u, ∆, A) recurrences.". This paragraph is a bit obscure. First, a definition of almost periodic function should be given at least as a reference (there are indeed several definitions, in the finite dimensional case considered here a simpler definition is possible). But most importantly the definition of almost period should be given. Moreover an almost period depends on an error \epsilon which should be made explicit. Finally clarifying the entire argument would greatly help the reader to follow the discussion.

10) Before Eq. (37): "The larger recurrences should have a duration comparable to that of the initial equilibration time T^A_{eq}, which we can define as the time it takes for eq ⟨A(t)⟩ to settle around the steady value ⟨A(∞)⟩."

This entire paragraph require addressing. First, it is not clear what it is meant with "larger recurrence". Are these the recurrences found for those u such that T_{recurrence}(u) is maximum? i.e. max_u T(u,∆,A) ?
But then an implication is presented with no explanation at all (the only argument is "should"). Why recurrences should have anything to do with the equilibration time (correctly defined here)? Clearly the equilibration time depends very strongly on the initial conditions (it only matters what happens at the beginning of the evolution) whereas the recurrence time, as defined here, is an average over many times very far away from the initial evolution. Also the "derivation" of Eq. (37) is completely arbitrary and I really find it hard to see any argument in the text. It seems that the goal of this discussion is to derive a time scale for the recurrence time. And for the observable they "argue" ∆ ∼ T_eq^A, while for the fidelity ∆ ∼ 1/σ. Once again there are some troubles connected with ∆ which seems to be a free parameter but not entirely.

Indeed, analogously, another argument is presented before Eq. (39) which "closely matches the behaviour found in [10]". Here the time scale would be given by 1/\sigma (\sigma^2 is the variance of the energy obtained with the initial state). To tell the truth this does not coincide with the time scale found in [10] as the standard deviation found there is obtained with a different state.

In any case it seems to me that the whole discussion around Eq. (37) and Eq. (39) is confusing, not precise, and frankly also potentially not necessary. The authors can probably still argue that their bound Eq. (38) is qualitatively similar to the result found in [10]. In summary my suggestion is either i) to find better, clearer arguments for the "derivation" of ∆ ∼ T_eq^A, and ∆ ∼ 1/σ (perhaps by finding another definition of recurrence time that does not require \Delta, which would remove the problem altogether), or ii) simply to remove the parts around Eqns. (37) and (39). In any case I mention the following point which I believe is a typo.

11) Between Eq. (38) and (39): "σ^2 = ⟨Φ|H^2 |Φ⟩ − ⟨Φ|H |Φ⟩^2 ". I believe |Φ⟩ should be the initial state which has always been indicated with \Psi so far.

12) After Eq. (38). "This closely matches the behaviour found in [10], which gives an exact calculation of the average recurrence time assuming a Gaussian wavefunction.". I don't think any particular assumption is made in ref [10] concerning the initial wavefuction. Certainly not Gaussianity (of which I would not know the meaning in this context). The only assumption there seems to be that of rational independence of the eigenvalues, which is slightly stronger than the assumption made here (of "generic spectrum").

13) In appendix A before using the dominated convergence theorem, to be rigorous I believe the authors should prove that the limit as T\to \infty in the definition of P(x) exists (i.e. that P(x) exists and is well defined).

14) In appendix C, before Eq. (C10), "The number of distinct derangements is denoted by !q, the subfactorial of q, and has no explicit formula.". I believe that !q = round(q!/e) is a quite explicit formula. Moreover the authors should show why !q <= q^q which should be easy to derive with this formula (something like q!/e < q! < q^q)

15) after Eq. (C25): "where c' is weakly...". there is no c' in Eq. (25) did the authors meant c?

16) Appendix D. "We choose pure states that allow us..", probably slightly clearer -> "We choose pure initial states that allow us...".

---

## Round 1 · Referee Report · Anonymous (Referee 2) · 2023-5-30

Report

The authors consider time-averaged relaxation after quantum quenches in isolated many-particle quantum systems. As argued by Srednicki in his seminal 1999 paper the ETH provides an estimate for the probability of finding the time-evolving expectation value of an observable at a fixed distance from its long time average. In their work the authors obtain a rigorous concentration bound for this probability. I think this is an important result in mathematical physics (even though infinite time averages in finite systems are a purely theoretical device) which warrants publication in SciPost Physics.
I have a couple of minor questions I would like the authors to consider .
(i) What does the requirement of free fermion Hamiltonians having a generic spectrum translate to in terms of the hopping amplitudes t_{n,m}? Clearly a simple tight-binding model will generally not fulfil this requirement. Is it obvious that an "infinitesimally" small deformation will lead to a Hamiltonian that has a generic spectrum?
(ii) I am somewhat puzzled by the definition of recurrences implied by (32) and (33) and in particular by having a finite value of \Delta. The naive expectation is that having a finite value of \Delta generically will preclude recurrences with very small values of u. The lower bound derived by the authors does not contradict this intuition. A much more physically relevant definition of a recurrence would refer to the limit \Delta\to 0, but then the bound becomes trivial. Can the authors comment on this?

---

## Round 2 · Referee Report · Anonymous (Referee 3) · 2023-8-3

Report

The authors have addressed the questions I raised in my first report in a satisfactory manner. I therefore recommend publication in SciPost Physics for the reasons given in my initial report.

---

## Round 2 · Referee Report · Lorenzo Campos Venuti (Referee 1) · 2023-9-5

Strengths

Same as previous

Report

The authors have satisfactorily answered all my comments. I therefore encourage publication of the manuscript.

Minor point/typo, here are still a few occurrences of <A(\infty)>.

---

## Editorial Decision

published